# Translating away Translationese without Parallel Data

**Rricha Jalota[1,3,*], Koel Dutta Chowdhury[1],**
**Cristina España-Bonet[2], and Josef van Genabith[1,2]**
[1]Saarland University, Saarland Informatics Campus, Germany
[2]German Research Center for Artificial Intelligence (DFKI)
[3]AppTek GmbH, Germany
rjalota@apptek.com, rrja00001@stud.uni-saarland.de
koel.duttachowdhury@uni-saarland.de
{cristinae, Josef.Van_Genabith}@dfki.de

## Abstract

Translated texts exhibit systematic linguistic differences compared to original texts in the same language, and these differences are referred to as translationese. Translationese has effects on various cross-lingual natural language processing tasks, potentially leading to biased results. In this paper, we explore a novel approach to reduce translationese in translated texts: translation-based style transfer. As there are no parallel human-translated and original data in the same language, we use a self-supervised approach that can learn from comparable (rather than parallel) mono-lingual original and translated data. However, even this self-supervised approach requires some parallel data for validation. We show how we can eliminate the need for parallel validation data by combining the self-supervised loss with an unsupervised loss. This unsupervised loss leverages the original language model loss over the style-transferred output and a semantic similarity loss between the input and style-transferred output. We evaluate our approach in terms of original vs. translationese binary classification in addition to measuring content preservation and target-style fluency. The results show that our approach is able to reduce translationese classifier accuracy to a level of a random classifier after style transfer while adequately preserving the content and fluency in the target original style.

## 1 Introduction

Translated texts often exhibit distinct linguistic features compared to original texts in the same language, resulting in what is known as *translationese*. Translationese has a tangible impact on various cross-lingual and multilingual natural language processing (NLP) tasks, potentially leading to biased results. For instance in machine translation (MT), during training, when the translation

direction of parallel training data matches the direction of the translation task, (i.e. when the source is original and the target is translated), MT systems perform better (Kurokawa et al., 2009; Lembersky et al., 2012). Similarly, Toral et al. (2018), Zhang and Toral (2019) and Graham et al. (2020) show that translating already translated texts results in increased BLEU scores. More recently, Artetxe et al. (2020a) observed that cross-lingual models when evaluated on translated test sets show false improvements simply due to induced translation artifacts. When investigating the effects of translationese in cross-lingual summarization, Wang et al. (2021) found that models trained on translated training data suffer in real-world scenarios. These examples show the importance of investigating and mitigating translationese. Despite this, removing translationese signals in already generated output of translations is an underexplored research topic. Dutta Chowdhury et al. (2022) remove translationese implicitly encoded in vector embeddings, and demonstrate the impact of eliminating translationese signals on natural language inference performance. Wein and Schneider (2023) leverage Abstract Meaning Representation (AMR) as an intermediate representation to abstract away from surface-level features of the text, thereby reducing translationese. Neither of these works explicitly analyzes surface forms of the resulting "debiased text" and its resemblance to original texts.

In this work, we aim to reduce the presence of translationese in human-translated texts and make them closely resemble original texts, notably, without using any parallel data as parallel human original - translated data in the same language do not exist. To this end, we explore a self-supervised neural machine translation (NMT) system (Ruiter et al., 2019) and its application to style transfer (ST) (Ruiter et al., 2022). In both works, validation is performed on a parallel dataset, either bilingual (MT data) or monolingual (ST data). However,

---

*Work done while the author was a student at Saarland University.

parallel human original-translationese data in the same language are unavailable. To overcome this challenge, we define an unsupervised validation criterion by combining a language model loss and a semantic similarity loss, inspired by Artetxe et al. (2020b). Our baseline is the self-supervised approach (SSNMT) from Ruiter et al. (2019). However, we go a step further and propose a novel joint training objective that combines both self-supervised and unsupervised criteria, eliminating the need for parallel data during both training and validation.

The contributions of this work are as follows:

- We are the first to formulate reduction of translationese in human translations as a monolingual translation based style-transfer task, allowing for direct evaluation of the effects on the surface forms of the generated outputs.

- We introduce a joint self-supervised and unsupervised learning criterion that eliminates the need for a parallel (and non-existent) original - translated dataset (in the same language) during training and validation.

- We show that our method is able to reduce the accuracy of a translationse classifier to that of a random classifier, indicating that our approach is able to successfully eliminate translationese signals in its output.

- We present an extensive evaluation that measures (i) the extent to which our methods mitigate translationese as well as adequacy and fluency of the outputs (**Quantitive Analysis**), (ii) estimates the degree of translationese in the output using metrics derived from linguistic properties of translationese (**Qualitative Analysis**).

## 2 Related Work

### 2.1 Text Style Transfer

Text style transfer is the task of altering the stylistic characteristics of a sentence while preserving the original meaning (Toshevska and Gievska, 2022). The amount of parallel data available for this task is usually limited. Therefore, readily available mono-stylistic data together with a smaller amount of style-labeled data are often exploited using approaches based on self- (Ruiter et al., 2019), semi- (Jin et al., 2019) or unsupervised Neural Machine Translation (Lample et al., 2018a; Artetxe et al., 2019). Common approaches involve disentangling the style and content aspects of the text. For content extraction, approaches based on variational auto-encoders (VAE) (Shen et al., 2017; Fu et al., 2017), cycle consistency loss (Lample et al., 2019), or reinforcement learning (Xu et al., 2018) are commonly employed. To induce the target style, often a style discriminator is employed using a pretrained style classifier (Prabhumoye et al., 2018; dos Santos et al., 2018; Gong et al., 2019) or the decoder head is specialized to generate target-style outputs (Tokpo and Calders, 2022), or the content representation is simply concatenated with the target-style representation (Fu et al., 2017).

Since unsupervised methods often perform poorly compared to their supervised counterparts (Kim et al., 2020; Artetxe et al., 2020b), recent approaches have explored semi-supervised (Jin et al., 2019) and self-supervised learning (Ruiter et al., 2022; Liu et al., 2022). Liu et al. (2022) combine sentence embeddings with scene graphs to mine parallel sentences on-the-fly for facilitating reinforcement learning-based style-transfer, while Ruiter et al. (2022) follow a simpler approach that exploits only the latent representations for online parallel sentence pair extraction from comparable data and leverage these pairs for self-supervised learning. Although their approach requires a parallel validation set for model selection and hyperparameter tuning, due to its simplicity, we adopt it as our starting point and baseline. We then present a novel version of this approach using unsupervised techniques, eliminating the need for a parallel validation set.

### 2.2 Unsupervised Model Selection

Several studies exploring unsupervised or semi/self-supervised settings either do not report their validation scheme (Artetxe et al., 2020b) or are only unsupervised (or semi/self-supervised) during training and rely on parallel in-domain validation sets for model tuning (Marie and Fujita, 2018; Marie et al., 2019; Dai et al., 2019; Ruiter et al., 2022). In contrast, some studies enforce strictly unsupervised settings in NMT by either using the validation set from a separate language pair (Artetxe et al., 2018a,b) or not using a validation scheme at all (Lample et al., 2018c), risking sub-optimal models. To address this, Lample et al. (2018b,a); Artetxe et al. (2020b) proposed using an unsuper-

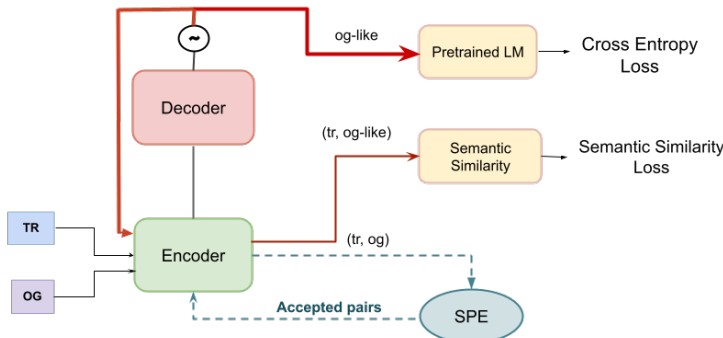

Figure 1: Model architecture. Here $(tr, og)$ is a (Translated (source), Original (target)) style sentence pair, and $og$-like is the translationese-mitigated output. The dashed arrows correspond to on-the-fly parallel pair extraction that facilitates Supervised Training, while the red arrows in bold represent the path of approximated decoder outputs used in Unsupervised Training.

vised validation criterion over monolingual data that conforms with the evaluation criterion or is guided by the target-distribution, similar to Artetxe et al. (2019) who combined cycle consistency loss with language model loss for unsupervised model selection.

### 2.3 Translationese Mitigation

Researchers have explored the effects and origins of translationese in previous studies. To mitigate translationse effects in machine translation (MT) models, a prevalent approach is tagged training (Caswell et al., 2019; Marie et al., 2020; Riley et al., 2020). This technique involves explicitly marking the translated and original data using tags, enabling the models to recognize and account for these distinctions. Yu et al. (2022) introduced an approach to mitigate translation artifacts in the `translate-train`[1] cross-lingual setting. To reduce translation artifacts in the target language, they learned an original-to-translationese mapping function from the source language. They do this by projecting the original and translated texts in the source language to a common multilingual embedding space and then learning to minimize the distance between the mapped representations of the originals and translationese. Dutta Chowdhury et al. (2022) tackle the reduction of translationese from a different perspective, treating it as a bias in translated texts. They employ a debiasing approach to mitigate translationese by attenuating it in the latent representations, while Wein and Schneider (2023) reduce translationese using AMR as intermediate representations. However, none of the above stud-

ies specifically analyze the surface forms of the "debiased text".

To date, to the best of our knowledge, monolingual translation based style transfer on translation outputs to mitigate translationese has not been explored. To some extent, this is expected, as, at least for monolingual translation-based style transfer, parallel original and translated texts in the same language do not exist. Below we present a novel approach that builds on translation-based style transfer but unlike previous work without parallel data for both training and validation sets.

## 3 Translationese Mitigation via Style Transfer

Our goal is to eliminate translationese signals from translated texts by transforming them into an original-like version. We define two style attributes, $og$ and $tr$, representing original style and translated style, respectively. Given a text sample $x$ belonging to $tr$, our aim is to convert this $x_{tr}$ to $x_{og}$, where $x_{og}$ belongs to style $og$ but retains the same semantic content as $x_{tr}$. We denote the corpus with original sentences $OG$ and the corpus with translated sentences $TR$. We illustrate the process in Figure 1.

### 3.1 Self-Supervised Architecture

In our work we build on a Transformer-based ENC–DEC self-supervised system that jointly learns sentence pair extraction and translation in a virtuous loop. Given two comparable mono-stylistic corpora ($OG$ and $TR$), a sentence-pair extraction (SPE) module (Ruiter et al., 2019) utilizes the internal representations of the sentence pairs to extract sentences with similar meanings. This similarity matching module employs two types of latent rep-

---

[1] In this setting, the training data is translated from the source language into the target language and the translated texts are used for training.

| Source [Translated] | Target [Original] |
| --- | --- |
| This is an area in which we need to press on. | This is another aspect we have to work on. |
| My group fully supports the substance of what you have said. | My group has discussed in detail the questions that you have posed. |
| That is not at all the case. | That is not the case at all. |
| I shall endeavour to be brief. | I will try to be brief. |

Table 1: Examples of accepted monolingual original-translationese pairs.

resentations: the sum of word embeddings $w(*)$ and the sum of encoder outputs $e(*)$. The embedded pairs ($\{w(tr), w(og)\}$ and $\{e(tr), e(og)\}$) are individually scored using a margin-based measure (Artetxe and Schwenk, 2019), and the top candidate pairs are selected for further filtering. Following Ruiter et al. (2019), we apply two filtering criteria:

- **Without Threshold [1]**: A sentence pair is accepted for training if it is highest-ranked in both candidate-pair representations. This is used in Ruiter et al. (2022).

- **With Threshold [2]**: A sentence pair is accepted for training either if it is highest-ranked in both candidate-pair representations or if its encoder representation $\{e(tr), e(og)\}$ surpasses a given threshold.[2]

Examples of extracted accepted pairs are shown in Table 1.

Extracted parallel sentence pairs are used in an online fashion to train the ENC–DEC model in a supervised manner by minimizing the cross-entropy loss ($L_{sup}$):

$$L_{sup} = -\sum_{j=1}^{N}\sum_{i=1}^{V} \boldsymbol{Y_i^j} \log(\boldsymbol{H_i^j}) \qquad (1)$$

where $N$ is the length of the target sequence, $V$ is the shared ENC-DEC vocabulary size, $\boldsymbol{Y_i^j}$ represents the $i$-th element of the one-hot encoded true distribution at $j$-th position and $\boldsymbol{H_i^j}$ the $i$-th element of the predicted distribution (hypothesis) $\boldsymbol{H_i}$. The joint SPE-translation learning loop continues until convergence. We use BART-style denoising autoencoding (DAE) (Lewis et al., 2020) for model initialization (see details in Section 4.2).

---

[2]The threshold was determined empirically by inspecting values in the range of $[0.95, 1.03]$, and validated through a manual assessment of the quality of accepted pairs. We observe that with a higher threshold, fewer yet better quality (tr,og) parallel pairs were extracted. We, therefore, set the value to 1.01 or 1.02 in our experiments (see A.5) as it provided a good trade-off between the quality and quantity of accepted pairs.

## 3.2 Joint Training Architecture

In the baseline system, all sentence pairs rejected by SPE are simply discarded, which is a major loss of useful mono-stylistic information. One way to utilize the discarded pairs is by combining the supervised training criterion with unsupervised training. To this end, we introduce an unsupervised loss component to the final objective, which combines language model (LM) loss with semantic similarity loss. Both the losses, as shown in Figure 1, are computed over the decoder outputs when mono-stylistic $tr$ is given as input.

As the input to compute these losses is derived from a categorical distribution (i.e. after applying argmax on the decoder output), this breaks the overall differentiability of the model during training. Therefore, following Yang et al. (2018) and Unanue et al. (2021), during training, we use continuous approximations of the decoder output to compute the two losses.

Let $y_{\hat{og}}$ be the style-transferred output when $x_{tr}$ is given as input. Using greedy decoding, the predictions from the decoder at the *j-th* decoding step can be expressed as follows:

$$\hat{y_{og}}_j = \arg\max_{x_{tr}} p(\hat{x_{og}}|x_{tr}, \hat{y_{og}}_{j-1}, \theta); j = 1\ldots, k \qquad (2)$$

To retain the end-to-end differentiability of the model, we replace the discrete argmax operation with a Gumbel-Softmax (Maddison et al., 2017; Jang et al., 2017) distribution and obtain continuous approximations of the decoder outputs. Let $p_j^i$ denote the probability of the *i-th* token in the $V$-sized vocabulary at the *j-th* decoding step in Equation 2 and $\boldsymbol{p}_j$ represent the entire probability vector at step $j$. Then the components of $\boldsymbol{p}_j$ can be approximated using:

$$\pi_j^i = \frac{\exp((\log p_j^i) + g^i)/\tau}{\sum_{v=1}^{V}\exp((\log p_j^v + g^v)/\tau)} \qquad (3)$$

where $g_i$ is a sample drawn from the Gumbel(0,1) distribution and $\tau$ is a temperature parameter[3] that controls the sparsity of the resulting vectors.

---

[3]set to 0.1 (Jauregi Unanue et al., 2021)

The continuous probability vectors denoted as $\boldsymbol{\pi}_j$ at each decoding step $j$, represent probabilities over tokens in the shared ENC–DEC vocabulary. During the training phase, we use these vectors to compute the language model loss and the semantic similarity loss. We define our language model loss and semantic similarity loss below.

**Language Model Loss**: To ensure target-style fluency, we follow the continuous-approximation approach from Yang et al. (2018). In this approach, we use the language model as a discriminator. The language model is initially pretrained on the target-style (i.e. the originals) to capture the target-style distribution and is denoted as $LM_{og}$. We feed the approximated decoder output $\boldsymbol{\pi}_j$ (from the $j$-th decoding step) to the pretrained language model $LM_{og}$ by computing the expected embeddings $E_{lm}\boldsymbol{\pi}_j$, where $E_{lm}$ represents the embedding matrix of $LM_{og}$. The output received from $LM_{og}$ is a probability distribution over the vocabulary of the next word, $\boldsymbol{q}_{j+1}^{og}$. Then the loss at $j$-th step is defined as the cross-entropy loss as shown below:

$$L_{lm} = -\boldsymbol{\pi}_{j+1} \log \boldsymbol{q}_{j+1}^{og} \qquad (4)$$

When the output distribution from the decoder $\boldsymbol{\pi}_{j+1}$ matches the language model output distribution $\boldsymbol{q}_{j+1}^{og}$, the loss achieves its minimum.

**Semantic Similarity Loss**: To enforce content preservation, the encoder representations of the input translation $e(tr)$ and the expected token embeddings from the decoder $e(E_{enc}\boldsymbol{\pi})$ are used to compute cosine similarity. Here, $E_{enc}$ represents the embedding matrix of the style transfer transformer encoder and $e(*)$ refers to the contextualized encoder representation. We define the loss as the mean-squared error of the cosine similarity loss:[4]

$$L_{ss} = \frac{1}{M} \sum_M (1 - \cos(e(x_{tr}), e(E_{enc}\boldsymbol{\pi})))^2 \quad (5)$$

where $M$ refers to the number of input sentences in a batch.

**Training and Validation**  To achieve a continuous approximation of the decoder output at each time-step and ensure end-to-end differentiability, we employ a two-pass decoding approach (Mihaylova and Martins, 2019; Zhang et al., 2019;

Duckworth et al., 2019). This approach works particularly well as we only feed mono-stylistic input to the Transformer (Vaswani et al., 2017) for unsupervised training. During training, the Transformer decoder is run once without accumulating gradients, and the (shifted) predicted sequence together with the encoder output are then fed into the Transformer decoder again to compute the unsupervised losses described above.

During the validation phase, the output from the first-pass decoding is used to compute the semantic similarity in terms of BERTScore between the input translation and the style-transferred output. Additionally, mean per-word entropy is computed over the style-transferred output. Note that, to measure semantic similarity between input $x_{tr}$ and output $y_{\hat{o}g}$ during training, the Style Transfer Encoder is used to compute cosine similarity while during validation, a pretrained BERT (Devlin et al., 2018) model is employed for measuring the BERTScore. The reason is that as the Style Transfer Encoder is continually learning to represent the two styles in every iteration, the pretrained BERT model provides a more stable and reliable similarity measure at every validation step.

The unsupervised objective is a linear combination of the language model loss and the semantic similarity loss, weighted by hyper-parameters $\beta$ and $\gamma$:[5]

$$L_{unsup} = \beta L_{lm} + \gamma L_{ss} \qquad (6)$$

The final loss is the sum of the supervised and unsupervised components:

$$L = \alpha L_{sup} + (1 - \alpha) L_{unsup} \qquad (7)$$

The hyperparameter $\alpha$ determines the balance between the two objectives and can either be fine-tuned or trained[6]. Note that while validation happens only over the unsupervised loss, in the final joint training objective, both the supervised $L_{sup}$ (Eq.1) and the unsupervised loss $L_{unsup}$ (Eq.6) are considered. Furthermore, in joint training, an initial self-supervised training phase is carried out for 300 optimization steps for English (EN) and 600 optimization steps for German (DE) to facilitate style-transfer learning in a guided manner and only then it is combined with unsupervised training.

---

[4]We also experimented with directly minimizing the cosine embedding loss at a lower learning rate and observed no differences.

[5]In our experiments, both $\beta$ and $\gamma$ are set to 1.

[6]We set it to 0.7 in our experiments to regulate the effect of weaker unsupervisory signals as monostylistic $TR$ data is much larger than the accepted pairs used for supervised learning.

## 4 Experimental Settings

### 4.1 Data

**Training Data**: We use a subset of the EuroParl corpus annotated with translationese information from (Amponsah-Kaakyire et al., 2021) (referred to as MPDE). Our focus is on two language scenarios: (i) **EN-ALL**: English originals (EN), and German and Spanish (ES+DE=ALL) translated into English and, (ii) **DE-ALL**: German originals (DE), and English and Spanish (ES+EN=ALL) translated into German.

**Validation and Test data:** The Self-Supervised baseline (SSNMT) relies on a parallel validation set for hyperparameter tuning. However, as this kind of data does not exist naturally, we generate machine-translationese (mTR) validation and test data by round-trip translating the original sentences (og) in the target language. Similarly, we denote the human-translationese as hTR. For our baseline SSNMT-based monolingual style transfer model, instead of using unaligned (hTR,og) pairs for validation, we utilise aligned (mTR,og) pairs.

For EN-ALL, we translate the original sentences from the EN–DE validation and test splits using Facebook's pre-trained WMT'19 EN→DE and DE→EN models (Ng et al., 2019) and for DE-ALL, we use M2M100 (Fan et al., 2021) with EN as the pivot langauge for round-trip translation. Refer to Appendix A.1 for the dataset statistics of the Style Transfer model.

For DAE and Language Model pretraining (for Joint Training), English and German monolingual data are collected from the EuroParl Corpus (Koehn, 2005) from the OPUS website[7] (Tiedemann, 2012). The English corpus contains over $1.5M$ train sentences and $5k$ dev and test sentences, while the German one has $2.1M$, $5k$, $5k$ train, test and dev sentences, respectively. Noisy input data for BART-style pretraining is generated with the values of parameters reported in Ruiter et al. (2022). For LM finetuning, we use the Original training split of the Comparable Dataset used for Style Transfer (see Appendix A.1).

### 4.2 Model Specifications

We implement our baseline system[8] and the proposed Joint-Training system[9] in Fairseq (Ott et al., 2019), using a *transformer-base* with a maximum

sequence length of 512 sub-word units. For online parallel-pair extraction, SSNMT requires indexing the mono-stylistic corpora for fast access. We use FAISS (Johnson et al., 2019) for this purpose. Sentence vectors are clustered into buckets using `IVF100` (wherein 100 equals $k$ in k-means[10]) and stored without compression (i.e. with `Flat` indexing). At search time, the top 20 buckets[11] matching a query are examined. All other parameters are set to the values reported in (Ruiter et al., 2022).

In Joint Training, we use the Fairseq implementation of a Transformer Decoder (Vaswani et al., 2017) Language Model to compute Language Model Loss during training and validation. The same model is used to measure perplexities during testing. Further information regarding the hyperparameters used for training the Style-Transfer models under different scenarios (i.e. with or without threshold across different data settings) can be found in Appendix A.5.

### 4.2.1 Classifier

The style transfer models are evaluated using a BERT-based (Devlin et al., 2018) binary classifier trained to distinguish human-translated data from originals. For English, we finetune `bert-base-cased` for the translationese classification task, and for German, `bert-base-german-cased`. The binary classifier is trained on the MPDE data with equal amounts of human-translated and original data (hTR,og). For EN-ALL, the training, validation and test splits for binary classification consist of 96536, 20654 and 20608 sentences, respectively while for DE-ALL, they consist of 87121, 18941 and 18976 sentences, respectively.

## 5 Evaluation

We perform evaluation on the outputs ($\hat{og}$) of the style transfer models, given the human-translationese (hTR) (without an original reference) half of the test data as input.

### 5.1 Quantitative Analysis

We compute three metrics: Acc. (Translationese Classification Accuracy), BERTScore (BERT-F1) and Perplexity (PPL)

**Acc.:** This metric measures the extent to which the models mitigate translationese, in terms of the

---

[7]http://opus.nlpl.eu/
[8]https://github.com/cristinae/fairseq/pull/4
[9]https://github.com/cristinae/fairseq/pull/5

[10]This value is set based on the size of our corpus and the recommendation in FAISS wiki.

[11]The higher the value, the longer the search time.

| Setup | Quantitative | | | Qualitative | | |
|---|---|---|---|---|---|---|
| **EN-ALL** | **Acc.[1]** | **Bert-F1** | **PPL$_{og}$** | **TTR** | **LD** | **#Identical** |
| Original Test $OG$ | | | | 0.902 | 33.70 | |
| Human-Translation Test | 79.62 | | 108.75 | 0.896 | 31.86 | |
| Self-Supervised **[1]** | 71.00 | **0.94** | 37.24 | 0.841 | 27.13 | 1186 |
| Self-Supervised **[2]** | 66.67 | **0.94** | **33.35** | 0.865 | 27.60 | 1186 |
| Joint Training **[1]**-hTR | 55.97 | 0.90 | 38.95 | 0.861 | 29.10 | 280 |
| Joint Training **[2]**-hTR | 57.32 | 0.90 | 50.98 | **0.887** | **32.33** | 390 |
| Joint Training **[1]**-mTR | **52.89** | 0.89 | 42.29 | 0.872 | 29.43 | 215 |
| Joint Training **[2]**-mTR | 57.26 | 0.90 | 50.91 | **0.887** | 32.32 | 390 |

(a) Results on the English MPDE test data from different training setups.

| Setup | Quantitative | | | Qualitative | | |
|---|---|---|---|---|---|---|
| **DE-ALL** | **Acc.[1]** | **Bert-F1** | **PPL$_{og}$** | **TTR** | **LD** | **#Identical** |
| Original Test $OG$ | | | | 0.919 | 33.49 | |
| Human-Translation Test | 79.30 | | 142.66 | 0.913 | 35.66 | |
| Self-Supervised **[1]** | 76.03 | 0.95 | **60.73** | 0.861 | 33.21 | 1913 |
| Self-Supervised **[2]** | 77.89 | **0.97** | 104.07 | **0.904** | 34.79 | 3481 |
| Joint Training **[1]**-hTR | 61.26 | 0.87 | 75.36 | 0.876 | 34.73 | 148 |
| Joint Training **[2]**-hTR | 63.50 | 0.87 | 82.47 | 0.843 | 39.75 | 181 |
| Joint Training **[1]**-mTR | **59.48** | 0.87 | 77.99 | 0.881 | 35.51 | 146 |
| Joint Training **[2]**-mTR | 59.92 | 0.84 | 100.41 | 0.796 | **54.05** | 141 |

(b) Results on the German MPDE test data from different training setups.

(c) Notation: No Threshold:[1], With Threshold:[2]; Validation set: human-translation(hTR)/machine-translation (mTR); Acc.[1] classification accuracy on the entire test set (hTR, og) and on the style-transferred outputs ($\hat{og}$, og); #Identical: #outputs that are same as the input. *Bold numbers highlight the best overall result under each metric while the second best results are underlined.*

accuracy of classifying the styles (translated vs. original). We report classification results on the entire test set (hTR, og) and on the test sets generated from the outputs of our style transfer models ($\hat{og}$, og). Note that, a near-random accuarcy indicates better style transfer.

**BERT-F1** is computed between the input translations and the style-transferred outputs to measure the degree of content preservation. For EN-ALL, we use pretrained `roberta-base` and for DE-ALL `xlm-roberta-base`.

**PPL** is calculated for the style-transferred outputs using a language model ($LM_{og}$) that has been fine-tuned on original text to evaluate the fluency and coherence of the target-style generated text.

For EN-ALL, Table 2a shows that the binary classifier achieves an accuracy of 79.62 on the (hTR, og) test set. This is our reference point. First, we study both self-supervised approaches — without [1] and with a threshold [2]. Results show that both the baseline approaches [1,2] were able to reduce translationese in human-translated texts while preserving the semantics (as evidenced by the F1-BERTScore of 0.94). More precisely, Self-Supervised [2] reduces the classification accuracy by $16\%$ when compared to the reference and by $11\%$ when compared to Self-Supervised [1]. Furthermore, the style-transferred outputs from Self-Supervised [2] show a reduction in $LM_{og}$ perplexity

when compared to the hTR texts and a $10.5\%$ reduction when compared to Self-Supervised [1] outputs. This indicates, for the baseline self-supervised approach, additionally relying on a threshold to extract more parallel pairs helps improve style transfer for EN-ALL MPDE[12].

Next, we examine if the same holds true for the Joint Training architecture and if joint training further improves the style transfer. In Table 2a, the consistent close-to-random accuracy across all the four variants of the Joint Training setup confirm the efficacy of this approach. The results show that in the Joint Training setup, regardless of the validation distribution (i.e., hTR or mTR), the style-transferred outputs from the models with no thresholds (i.e. Joint Training [1]-hTR and Joint Training [1]-mTR) achieve $2\% - 8\%$ reductions in accuracy w.r.t. their counterparts with thresholds.

This observation suggests that when the model is trained with a larger amount of data through unsupervised training, it may be adequate to utilize only high-quality SPE pairs, without the need for including additional sub-optimal pairs based on a threshold. When comparing these two models against each other (i.e. Joint Training [1]-hTR vs. Joint Training [1]-mTR ), the generations from the model validated on hTR exhibit similar semantic

---

[12]See Appendix A.2.1 for the statistics on the accepted pairs.

similarity but with approximately $8\%$ lower perplexity, which suggests a potential advantage in employing the same distribution (hTR) for validation as during training and testing.

We replicate the same analysis for the **DE-ALL setup**, as demonstrated in Table 2b. Note that, this dataset is even smaller than the EN-ALL MPDE dataset (see Table 4). Here, the reference accuracy is 79.30 on (hTR, og) test set. When using the baseline Self-Supervised methods [1,2], the classification accuracy reduces only marginally by $2\%$ to $4\%$, with no significant distinction between the two variants. Self-Supervised [1], however, unlike in EN-ALL, benefits from a $42\%$ lower $LM_{og}$ perplexity than Self-Supervised [2]. Nonetheless, similar to the EN-ALL setup, with Joint Training, the degradation is more pronounced. In contrast to EN-ALL, however, the style-transferred outputs from Joint Training without threshold (i.e. Joint Training [1]-hTR and Joint Training [1]-mTR) exhibit only a marginal decrease in accuracy compared to their counterparts with threshold. Assessing content preservation and fluency in the target style within the Joint Training setup, all the models, except Joint Training [2]-mTR, yield similar results. Notably, the outputs from Joint Training [1]-hTR demonstrate the lowest perplexity. In case of Joint Training [2]-mTR, the oddly high perplexity when evaluating with $LM_{og}$ could be attributed to either the sub-optimal parallel pairs accepted by additionally applying a threshold or performing validation on mTR or both.

Finally, we compare the Joint Training setup with the baseline Self-Supervised setup across both the MPDE datasets. Overall, Joint Training reduces the classification accuracy to a near-random accuracy, with a more pronounced drop in accuracy for EN-ALL MPDE [13]. However, we also observe a degradation in the F1-BERTScore as we move to the Joint Training setup. This discrepancy can be attributed to the fact that Self-Supervised training yields higher number of outputs that are identical to the given input translations[14]. Therefore, a higher F1-BERTScore does not necessarily mean that the non-identical outputs from Self-Supervised base-

lines preserve the content better. Earlier studies have shown that BERTScore is sensitive to both meaning and surface changes (Hanna and Bojar, 2021; Zhou et al., 2022), and therefore, one needs to manually examine the outputs.

## 5.2 Qualitative Analysis

Prior research (Baker et al., 1993) has shown that translated texts are often simpler than original texts. In order to measure the level of translationese in a translation in terms of lexical simplicity, Toral (2019) introduced two metrics: Type-Token Ratio (TTR) and Lexical Diversity (LD). Following Riley et al. (2020), we conduct a qualitative analysis using these metrics.

**TTR**: Measures lexical variety by dividing the number of unique word types in a text by the total number of words (tokens) in the same text. The lower the ratio, the more reduced the vocabulary of the style-transferred output, indicating weak resemblance to the target original style.

**LD**: Calculated by dividing the number of content words (words that carry significant meaning - adjectives, adverbs, nouns and verbs) by the total number of words in the text. Higher content word density implies a text conveys more information, aligning it more closely with the original style.

Table 2a shows that for EN-ALL MPDE, the TTR and LD scores for the outputs of the baseline Self-Supervised approach [2] are higher than that of Self-Supervised [1]. This is in line with the quantitative results, which clearly indicate that additionally applying a threshold to retrieve a higher number of parallel pairs helps improve style transfer. However, this does not hold for Joint Training. The TTR and LD scores for Joint Training [2]-(hTR/mTR) models are higher than for the models without a threshold, although they achieve higher $LM_{og}$ perplexities. Interestingly, the LD scores from Joint Training [2]-hTR/mTR even surpass the reference LD score on hTR.

In case of DE-ALL MPDE (Table 2b), which has an even smaller size of training data, a lack of correlation between the quantitative and qualitative results is even more evident within the baseline self-supervised and the joint training setups. This suggests the need for an extrinsic evaluation on a downstream task, similar to the one performed by Dutta Chowdhury et al. (2022) on the Natural Language Inference (NLI) task.

In Table 3, we present some examples from the different variants of our Style Transfer system and

---

[13]To closely inspect the impact of style transfer, in Appendix A.2, we report the classification accuracies on only the translationese half of the test set. This provides an insight into the number of sentences in hTR or $\hat{og}$ that are considered by the classifier as original-like (see #OG-like in Table 6c).

[14]Recall that the target reference is unavailable, and BERTScore is computed between the translated input and the style-transferred output.

| Source (hTR) | Self-Supervised[2] | Joint Training[1]-mTR | Joint Training[1]-hTR |
|---|---|---|---|
| I happily leave it to you to examine this matter. | I leave it to you to examine this matter. | I will leave it to you to reflect on this matter. | I will therefore leave it to you to consider this matter. |
| Unfortunately, this hope has not become reality. | Unfortunately, this has not become reality. | Despite that, it has not become reality. | The reality is, however, rather different. |
| Please could he just explain that? | Could he just explain that? | Could he just explain that? | Could he please explain that? |
| Please could you take suitable action here. | Could you take suitable action? | Can you please do something about it? | Could you please take some action? |
| I am in favour of Turkey's admittance to the EU, but the Copenhagen criteria must be met. | I am in favour of Turkey's's admittance to the EU, but the Copenhagen criteria must be met. | I am very much in favour of Turkey's admittance to the EU, but these must be taken into account. | I am very much in favour of Turkey's accession to the European Union, but the Copenhagen criteria must be met. |

Table 3: Qualitative analysis of the outputs from different systems.

broadly analyse the generated outputs ($\hat{o}g$) with respect to some well-known linguistic indicators of translationese (Volansky et al., 2015)[15]. In the first and second example, the intended meaning and surface form is retained in the outputs from all systems while the outputs obtained from Joint Training[1]-hTR introduces a higher level of formality, altering both the connective and the phrase. In the second example, Self-Supervised[2] retains key elements from the source text, while Joint Training brings about more significant changes in terms of connectives, lexical choices, and sentence structure. In the third example, while Self-Supervised[2] and Joint Training[1]-mTR change the formulation of the question while removing politeness ("please"), Joint Training[1]-hTR keeps it intact. Similarly, in the fourth example, all three versions use different connectives, with varying politeness and formality. In last example, Self-Supervised retains a substantial portion of the source text, while Joint Training[1]-mTR eliminates the specific mention of the "Copenhagen criteria" and replaces it with "these," which is a more general reference and Joint Training[1]-hTr changes the phrase by using "accession to the European Union" instead of "admittance to the EU."

Overall, we observe that our proposed approach is able to mitigate translationese in its style-transferred outputs while preserving the semantics reasonably well and achieving greater fluency in target original style.

## 6   Conclusion

In this work, we reduce the presence of translationese in human-translated texts and make them more closely resemble originally authored texts. We approach the challenge as a monolingual

translation-based style transfer task. Due to the absence of parallel translated and original data in the same language, we employ a self-supervised style transfer approach that leverages comparable monolingual data from both original and translated sources. However, even this self-supervised approach necessitates the use of parallel data for the validation set to facilitate the learning of style transfer. Therefore, we develop a novel joint training objective that combines both self-supervised and unsupervised criteria, eliminating the need for parallel data during both training and validation. Our unsupervised criterion is defined by combining two components: the original language model loss over the style-transferred output and the semantic similarity loss between the input and style-transferred output. With this, we show that the generated outputs not only resemble the style of the original texts but also maintain semantic content. We evaluate our approach on the binary classification task between original and translations. Our findings show that training with joint loss significantly reduces the original and translation classification accuracy to a level comparable to that of a random classifier, indicating the efficacy of our approach in mitigating translation-induced artifacts from translated data. As future work, we intend to explore more sophisticated approaches to improve content preservation and evaluation. It would be also interesting to apply a version of our style transfer approach to the output of machine translation to alleviate translationese in cross-lingual scenarios.

## Limitations

The availability of original and professionally translated data in the same language, domain and genre is limited, both in terms of quantity and language coverage. While our system does not rely on par-

---

[15]See Appendix A.3 for the analysis of German outputs.

allel data, for our approach to work it is important to ensure that the data in both modalities (original and translationese) are comparable.

Manually evaluating the quality of style transfer and the reduction of translationese is inherently subjective. While we conduct a preliminary analysis to evaluate the outputs, more nuanced linguistic expertise is required for in-depth analysis. Though we propose evaluation metrics, there is no universally accepted gold standard for measuring the effectiveness of translationese reduction. These factors may introduce biases and challenges in comparing the performance of different approaches. At the individual text level, even human experts struggle to distinguish between translationese and originals. Detecting translationese reliably involves analyzing large quantities of data or training classifiers on original and translated text.While Amponsah-Kaakyire et al. (2022) show evidence that high-performance translationese classifiers may in some cases rely on spurious data correlations like topic information, recent work by Borah et al. (2023) indicates that the effect arising from spurious topic information accounts only for a small part of the strong classification results. A decrease in classifier accuracy strongly suggests reduced translationese signals. That said, further research on addressing spurious correlations in translationese classification is an important research not explored in our paper.

Finally, it is worth noting that the proposed systems may attempt to improve translations already of high quality that should not really be touched. Our results include potential quality degradation due to system overcorrections. Future work aims to address this phenomenon using an oracle-based style transfer approach.

## 7 Acknowledgments

We would like to thank Etienne Hahn for helping with the manual analysis of German outputs. We also thank the anonymous reviewers for their invaluable feedback. This work was funded by the Deutsche Forschungsgemeinschaft (DFG, German Research Foundation) – SFB 1102 Information Density and Linguistic Encoding.

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

## A Appendix

### A.1 Dataset Statistics for Style Transfer System

**Data Preprocessing:** First, the paragraphs are split into sentences using NLTK (Bird et al., 2009) and then tokenized and truecased using standard Moses scripts (Koehn et al., 2007). In addition, we remove all duplicates from the train/test/dev splits. After this, byte-pair encoding of 10k merge operations is applied on the concatenated monolingual-EN EuroParl and MPDE's EN-ALL training split and similarly, 11k merge operations are applied on the concatenated monolingual-DE EuroParl and the DE-ALL training split.

### A.2 Supplementary Results

### A.2.1 Parallel Pairs for Joint Training

Table 5 provides an insight on how the accepted parallel pairs influence style transfer training. The models corresponding to the underlined numbers

| Data | Split | OG | TR |
|------|-------|------|------|
| **EN-ALL** | train | 96,290 | 96,206 |
| | dev | 10,327 | 10,327 |
| | test | 10,304 | 10,304 |
| **DE-ALL** | train | 64,917 | 43,560 |
| | dev | 9,470 | 9,470 |
| | test | 9,488 | 9,488 |

Table 4: *Number of sentences in each split of the Original [OG] and Translationese[TR] halves of the MPDE dataset.*

attain the best checkpoint earlier than their counterparts and hence, we report the number of parallel pairs from this epoch. As seen in the quantitative analysis for EN-ALL, the large number of accepted pairs due to the application of a threshold in the self-supervised approach, helps improve style transfer. However, for the Joint Training, due to the weaker unsupervised signals, the threshold does not have the same impact. Interestingly, for DE-ALL, the additional use of a threshold in both the Self-Supervised and Joint Training approaches does not increase the number of parallel pairs. Furthermore, across both the data setups, Joint Training with no threshold achieves a greater number of parallel pairs than its counterpart with threshold. This explains why the quantitative metrics perform slightly better for this model variant.

| Data | Setup | Epoch | No Thresh. | Thresh. |
|------|-------|-------|------------|---------|
| EN-ALL | Self-Supervised | 64 | 317 | 2,978 |
| | Joint Training-hTR | 2 | 492 | 405 |
| | Joint Training-mTR | 2 | 492 | 405 |
| DE-ALL | Self-Supervised | 12 | 416 | 362 |
| | Joint Training-hTR | 48 | 673 | 407 |
| | Joint Training-mTR | 48 | 673 | 398 |

Table 5: Statistics of the accepted parallel pairs for style transfer training. The number of parallel pairs are reported from the earliest epoch that gives the best model checkpoint between the two model variants: with and without threshold. In each row, the epoch number is associated to the underlined model.

### A.2.2 Translationese Classification on the Style-Transferred Outputs

In order to directly witness the impact of style transfer, we measure the accuracy just on the translationese half of the test set. This provides an insight into the number of sentences in the translated half that are considered by the classifier as original-like (see #OG-like in Table 6c). Note that, the lower the accuracy, the better the style transfer.

### A.3 Manual Inspection of German outputs

In Table 7, we provide examples from the style transfer models trained on MPDE DE-ALL. For

| EN-ALL | Acc.[2] | #OG-like |
|--------|---------|----------|
| Human-Translation Test | 82.83 | 1816 |
| Self-supervised [1] | 66.72 | 3430 |
| Self-supervised [2] | 58.87 | 4239 |
| Joint Training [1]-hTR | 35.70 | 6625 |
| Joint Training [2]-hTR | 38.40 | 6347 |
| Joint Training [1]-mTR | 29.54 | 7260 |
| Joint Training [2]-mTR | 38.28 | 6360 |

(a) Size of the test set: 10304.

| DE-ALL | Acc.[2] | #OG-like |
|--------|---------|----------|
| Human-Translation Test | 82.14 | 1695 |
| Self-supervised [1] | 76.15 | 2263 |
| Self-supervised [2] | 79.87 | 1908 |
| Joint Training [1]-hTR | 19.90 | 7600 |
| Joint Training [2]-hTR | 24.58 | 7156 |
| Joint Training [1]-mTR | 16.18 | 7953 |
| Joint Training [2]-mTR | 17.10 | 7866 |

(b) Size of the test set: 9488.

(c) Supplementary results on MOTRA EN-ALL and DE-ALL test sets. Notation: No Threshold:[1], With Threshold:[2]; Validation set: human-translation(hTR)/machine-translation (mTR); Acc.[2] classification accuracy on the translationese half of the test set (hTR,) and on the style-transferred outputs ($\hat{og}$,); #OG-like: #sentences in (hTR,) and ($\hat{og}$,) that are classified as original by the classifier.

the first two short sentences, although Joint Training[1]-hTR makes a different lexical choice with the use of "Klarstellung" and "allerdings", it does not alter the meaning of the two sentences. In the third example, with the use of "vertrauen darauf", Joint Training[1]-hTR heightens the intensity of the sentence while preserving the intended meaning. In the final example, the use of the pronoun "Dies" (*this*) by Joint Training[1]-mTR results in a loss of specificity regarding the city of Straßburg. However, both Self-Supervised[2] and Joint Training[2]-hTR manage to maintain the source sentence's meaning, albeit with a slight shift in the intensity of the word "schöne" (*beautiful*).

### A.4 Experimental Setup

We run our experiments on SLURM cluster using GPU instances of V100-32GB or RTXA6000. Both the baseline and the joint training approach are run on a single GPU while for DAE pretraining and LM training (pretraining and finetuning), we use 2 GPUs with 16 CPUs with each CPU containing 12 GB memory. One training on our datasets takes approximately 4 hours with the Baseline Approach and 7-9 hours for the Joint Training Approach.

Additionally, obtaining results for all the setups in the quantitative measures simultaneously takes approximately 30 minutes for each setup. How-

| Source (hTR) | Self-Supervised[2] | Joint Training[1]-mTR | Joint Training[1]-hTR |
|---|---|---|---|
| Das bedarf einer Erklärung . | Das bedarf einer Erklärung . | Das bedarf einer Erklärung . | Das bedarf einer Klarstellung |
| Es besteht jedoch ein Problem . | Es besteht jedoch ein Problem . | Es besteht allerdings ein Problem . | Es besteht allerdings ein Problem . |
| Wir hoffen, daß Sie es schaffen. | Wir hoffen, daß Sie es schaffen. | Wir hoffen, daß Sie es schaffen. | Wir vertrauen darauf, daß Sie es schaffen. |
| Straßburg ist durchaus eine schöne Stadt. | Straßburg ist eine schöne Stadt. | Dies ist eine schöne Stadt. | Straßburg ist eine sehr schöne Stadt. |

Table 7: Qualitative analysis of the German outputs from different systems.

ever, if we run them independently, it takes between $2 - 10$ minutes to obtain results using these measures. The qualitative measures, on the other hand, take longer to process due to the utilization of large spaCy models (`en_core_web_trf` for EN and `de_dep_news_trf` for DE), requiring approximately 15-20 minutes.

## A.5 Hyperparameters

SSNMT requires indexing the mono-stylistic corpora for fast access. We use FAISS (Johnson et al., 2019) for this purpose. For indexing with FAISS, since each half of the baseline corpus (details in Section 4.1) contains less than 1M sentences, to facilitate fast and accurate search, the sentence vectors are clustered into buckets using `IVF100` (wherein $100^{16}$ equals $k$ in k-means) and stored without compression (i.e. with `Flat` indexing). At search time, the top 20 buckets matching query are examined.

Other hyperparameter settings for the Style Transfer models are shown in Table 8.

The BERT-based Translationese binary classifiers are fine-tuned for at most 10 epochs with a learning rate of 2e-5, 1000 warm-up updates and a batch size of 16. The Transformer-based LM is trained for at most 50k updates with a learning rate of 5e-3 and fine-tuned for at most 20k updates at a lower learning rate of 5e-4 with the `eos` `sample-break-mode`.

The BERT-based Translationese binary classifiers are finetuned for at most 10 epochs with a learning rate of 2e-5 and 1000 warm-up updates. The Transformer-based LM is trained for at most 50k updates with a learning rate of 5e-3 and finetuned for at most 20k updates at a lower learning rate of 5e-4.

| Experiment | MPDE | Experimental Setting |
|---|---|---|
| Self-Supervised | | Learning Rate 0.0003 Validation Batch size 500 Warm-up updates 300 Batch-size 80 Save-interval-updates 2 Threshold 1.01 (when set) Epochs 100 |
| Joint Training | EN-ALL | Learning Rate 0.0003 Validation Batch size 500 Warm-up updates 300 Batch-size 160 Save-interval-updates 2 Threshold 1.01 (when set) Start-unsupervised training 300 Supervised-loss coefficient 0.7 Unsupervised-loss coefficient 0.3 Epochs 30 Patience 15 |
| Joint Training | DE-ALL | Learning Rate 0.0003 Validation Batch size 500 Warm-up updates 600 Batch-size 40 Save-interval-updates 2 Threshold 1.02 (when set) Start-unsupervised training 400 Supervised-loss coefficient 0.7 Unsupervised-loss coefficient 0.3 Epochs 30 Patience 15 |

Table 8: Hyperparameters used in the Style Transfer experiments.

---

[16]This value is set based on the recommendation in FAISS wiki.