# OpenReview forum: "Translating away Translationese without Parallel Data"
_EMNLP/2023/Conference — EMNLP 2023 Main_

### Official Review · Reviewer_whSa · 2023-07-26

**Soundness:** 4

**Excitement:**

4: Strong: This paper deepens the understanding of some phenomenon or lowers the barriers to an existing research direction.

**Missing References:**

Not a missing reference, but the opposite: (Vaswani et al., 2017a) and (Vaswani et al., 2017b) are the same.

**Paper Topic And Main Contributions:**

The paper addresses the issue of "translationese", the common problem found in translated text whereby the text exhibits linguistic features distinct from those of natively-written texts, often being closer to the features of the source language.

The authors tackle the task of correcting, or at least mitigating, the translationese present in translated text. The proposed approach is based on machine translation and style transfer techniques and works directly over the translationese text, changing it into text closer to one written natively in the target language.

Machine translation and style transfer techniques are typically trained and validated on parallel corpora, but there is no original-translationese parallel corpus that can be used for the task at hand. As such, the authors propose an unsupervised validation criterion based on the language model loss and the semantic similarity loss. The language model loss measures the fluency of the fixed translationese, as assessed by a language model for the target language, while the semantic similarity loss measures whether the semantic content has been preserved by comparing the similarity between the encoder representations of the translationese and the encoder representations of the fixed translationese.

Evaluation addresses multiple aspects:
- Fluency: The perplexity of the fixed translationese according to a language model for the target language.
- Content preservation: The BERTScore that measures the semantic similarity between the translationese and the fixed translationese.
- A sort of "adversarial" task: The accuracy of a translationese detector, which is expected to perform at about 50% (random choice) if the translationese has been successfully fixed.
- Finally, the type-token ratio and lexical diversity of the produced text are also measured.

The approach is novel, the method avoids the need for parallel corpora, and the evaluation shows that the goal of reducing the accuracy of a traslationese detector to random chance is achieved.

**Questions For The Authors:**

- A) For the self-supervised part (Section 3.1), you use the SPE module to pick matching sentences for tr-og pairs that will be used to train the encoder-decoder. Each sentence is tested against every other sentence? What is this SPE module, more concretely? Is it a BERT-like encoder? Did you use the one from (Ruiter et al., 2019) or did you train your own? On what corpora was SPE trained on?
- B) The SPE with threshold uses a threshold of 1.01. Can you say more on how this value was determined?
- C) When calcutating the probability vectors, a temperature value of 0.1 is used. Can you say more on how this value was picked?

[all questions were addressed in the rebuttal]

**Reasons To Accept:**

Unsupervised approaches seem almost magical, getting results from apparently nothing. This paper goes a bit along those lines, being able to fix translationese without any explicit examples (parallel corpora) of how to do it.

The following contributions that the authors chose to highlight are all good reasons to accept the paper:
- Tackling the task using a style-transfer approach is novel.
- The method combines self-supervised and unsupervised learning, eliminating the need for parallel corpora, corpora that doesn't even exist for this particular task.
- Translationese is successfully eliminated (according to an automatic classifier).

**Reasons To Reject:**

In my view, the main weaknesses of the paper concern the evaluation. I'll address these issues next.

Is the approach valid? Fluency is assessed by a language model, content preservation is assessed by an encoder, and accuracy is assessed by whether a translationese classifier can be "fooled" into random classification. This is fine for an automatic evaluation, but these are only proxies for the "real thing", which is whether the fixed translationese is natively fluent, preserves content, and is enough to "fool" a human. This has to be evaluated manually. The task proposed in this paper is begging for some sort of A/B testing where the translationese text and the result of fixing it are put side-by-side and presented to human translators, which then have to pick the best. Without this, the approach isn't sufficiently validated. This issue is partly alluded to in the Limitations section at the end of the paper, but it nevertheless remains as a weakness.

Is there the danger of this approach ruining translations that are already good? The proposed approach appears to be a great technique to improve automatic translations, so much so that it seems that it would be something that one would always want to run over the output of an automatic translation system. However, this raises the question of whether doing so can actually do more harm than good when the translated text is already of good quality and has little to no translationese. However, we don't have an oracle that, for a given translated text, can tell us if we're in this situation, and thus if we should or shouldn't attempt fix the translation. The paper makes no mention of this scenario.

The changes caused by this approach can go beyond simply fixing translationese. In Section 5.2, Table 4, the authors show some examples. In the fourth example, "unless radical changes are made" is transformed into "if this House goes along with". The authors show this as being a "sentence structure" change and state that it "[provides] a different perspective on the conditional statement". This transformation goes far beyond a structural/syntactic change or merely providing a differente perspective. The paper quickly glosses over this case, but a more detailed qualitative analysis of such results is in order.

[all issues were addressed in the rebuttal]

**Reproducibility:**

4: Could mostly reproduce the results, but there may be some variation because of sample variance or minor variations in their interpretation of the protocol or method.

**Reviewer Confidence:**

3: Pretty sure, but there's a chance I missed something. Although I have a good feel for this area in general, I did not carefully check the paper's details, e.g., the math, experimental design, or novelty.

**Typos Grammar Style And Presentation Improvements:**

- 077: "bilingual (MT) or monolingual (ST)": This is a bit confusing. Are the acronyms meant to refer to the techniques mentioned earlier? If so, it would be clearer to write something like "bilingual (using NMT) or monolingual (using ST)".
- p.3: The caption of Figure 1 is missing the closing parenthesis. In addition, it would be best if in Figure 1 and its caption you use curly brackets to denote pairs, as you've done in the body of the paper.
- 449: "Transfomer"
- 533: "translation. to rThis"
- 538: "Bert-score-based": Change to "BERTScore-based", to be consistent with the way you've written it elsewhere.
- p.5, Equation 5: It seems to me that the semantic similarity loss you show here is the cosine one, and not the MSE one.
- There are a few places (306, 438, 441) where you have a footnote reference right after a number, which is needlessly confusing as it makes it look like an exponent. I'd move the footnote reference to a nearby word.
- Section 5.2 concerns the qualitative analysis, but I'd consider the TTR and LS metrics that you use to still be a quantitative analysis, not qualitative. The closest thing to a qualitative analysis is the manual analysis of the output.
- Throughout the paper, I feel a term like "native" would be clearer and more explicit than "original". For instance, "Our goal is to eliminate translationese signals from translated texts by transforming them into an native-like version" (Section 3).

---

> ### Author Rebuttal · Authors · 2023-08-28
>
> We thank Reviewer 3 for the insightful feedback and comments.
>
> > Is the approach valid? Fluency is assessed by a language model, content preservation is assessed by an encoder, and accuracy is assessed by whether a translationese classifier can be "fooled" into random classification. This is fine for an automatic evaluation, but these are only proxies for the "real thing", which is whether the fixed translationese is natively fluent, preserves content, and is enough to "fool" a human. This has to be evaluated manually. The task proposed in this paper is begging for some sort of A/B testing where the translationese text and the result of fixing it are put side-by-side and presented to human translators, which then have to pick the best. Without this, the approach isn't sufficiently validated. This issue is partly alluded to in the Limitations section at the end of the paper, but it nevertheless remains as a weakness.
>
> **(Evaluation-Soundness/Validity)**  We agree that in an ideal scenario, A/B testing using human-expert evaluation should be conducted. However, translationese effects for professional human translation (as we are considering in the MPDE Europarl data ([Amponsah et. al (2021)](https://aclanthology.org/2021.motra-1.1)) are subtle, and even human experts are usually not able to reliably distinguish between translationese and originals reliably above chance ([Tirkkonen-Condit et. al (2002)](https://api.semanticscholar.org/CorpusID:144823252), [De Sutter et. al (2020)](https://www.tandfonline.com/doi/abs/10.1080/0907676X.2019.1611891)). In fact, to reliably detect translationese one either (i) has to collect sufficiently large quantities of original and translation data and do careful corpus analysis to detect reliable differences in the distributions of translationese relevant linguistic phenomena such as for example the distributions of discourse connectives suggesting “explicitation” in translationese or (ii) has to train classifiers on original and translated data ([Baroni et. al (2006)](ttps://api.semanticscholar.org/CorpusID:11006810), [Kurokawa et. al (2009)](https://aclanthology.org/2009.mtsummit-papers.9), [Rabinovich et. al (2015)](https://aclanthology.org/Q15-1030), [Volansky et. al (2015)](https://api.semanticscholar.org/CorpusID:5133943), [Pylypenko et al. (2021)](https://aclanthology.org/2021.emnlp-main.676)). Given enough training data and given enough data at prediction time, translationese classifiers can be astonishingly accurate, with up to > 90% accuracy ([Rabinovich et. al (2015)](https://aclanthology.org/Q15-1030), [Pylypenko et al. (2021)](https://aclanthology.org/2021.emnlp-main.676),[Amponsah et al. (2022)](https://aclanthology.org/2022.blackboxnlp-1.23)). In our work, we use SOTA BERT-based approaches as e.g. by [Pylypenko et al. (2021)]. Given that our data, is fairly uniform in terms of genre and domain (EU MPDE data), we (i) expect the classifiers to be reliable and (ii) reduce classifier accuracy to (near) random (per classification datum/input) to be a strong indication that (any previous) translationese signals have been strongly reduced in the data under classification. See also Table A for Reviewer 2. In this regard, the methodology and approach we follow are as good and as bad (as sound) as the current SOTA in the translationese research field at large ([Pylypenko et al. (2021)](https://aclanthology.org/2021.emnlp-main.676), [Dutta Chowdhury et al. (2022)](https://aclanthology.org/2022.naacl-main.292)).
>
> In addition to classification, we also use BERTScore as a proxy for content preservation and perplexity given a language model trained on original data as a proxy for closeness (or otherwise) to original data. Just as the classification results, these are proxies, reflecting current practice and SOTA in the field.
>
> In the revised version of the paper, we intend to compile a more extensive list of transformation examples (see Table C below) for an extended manual inspection and error analysis.
>
> > Is there the danger of this approach ruining translations that are already good? The proposed approach appears to be a great technique to improve automatic translations, so much so that it seems that it would be something that one would always want to run over the output of an automatic translation system. However, this raises the question of whether doing so can actually do more harm than good when the translated text is already of good quality and has little to no translationese. However, we don't have an oracle that, for a given translated text, can tell us if we're in this situation, and thus if we should or shouldn't attempt fix the translation. The paper makes no mention of this scenario.
>
> **(Soundness)** Yes, overcorrection happens, which is also an important point in research on Automatic Post-Editing (APE) in Machine Translation where the output of neural machine translation systems in high-resource training scenarios is already very good and changes required to make outputs even better are often subtle and risk corrupting already good translations. At the moment, all the results in the paper are inclusive of any deteriorations brought about by overcorrections on the part of our system. We will highlight this explicitly in the paper.
>
> Inspired by the reviewer’s question and their mentioning of an “oracle”, in the future, we may be able to extend the system architecture to incorporate oracle-guided style transfer. In this scenario, the oracle would be approximated by a pre-trained high-accuracy translationese classifier. During the training phase of the extended system, the decoder would receive two inputs: the extent of translationese in the input translated text (TR) (as inferred by the classifier) and the TR-encoder representations. This may enable the overall system to make a better-informed decision about style transfer based on the extent of translationese predicted in the translated input text. We will include this in the further work section.
>
>
> > The changes caused by this approach can go beyond simply fixing translationese. In Section 5.2, Table 4, the authors show some examples. In the fourth example, "unless radical changes are made" is transformed into "if this House goes along with". The authors show this as being a "sentence structure" change and state that it "[provides] a different perspective on the conditional statement". This transformation goes far beyond a structural/syntactic change or merely providing a differente perspective. The paper quickly glosses over this case, but a more detailed qualitative analysis of such results is in order.
>
> We agree: the example in Table 4 goes well beyond structural or syntactic change and we will discuss this better in the paper. The system transformed "unless radical changes are made" to "if this House goes along with", both articulating perhaps a negative stance towards the Buitenweg report, capturing the context of the source sentence as part of its broader implication, while abstracting away from the specific condition mentioned in the source sentence. We will adjust the discussion of this example accordingly. Below in Table C we provide further qualitative examples (hypotheses) across different versions of our approach:
>
> *Table C: Qualitative analysis of outputs from different systems.*
>
> | Source (tr) | Hypothesis1: Self-Supervised[2] | Hypothesis2: JointTraining[1]-mTr | Hypothesis3: JointTraining[1]-hTr |
> | ----------- | ------------------------------ | --------------------------------- | --------------------------------- |
> | I happily leave it to you to examine this matter. | I leave it to you to examine this matter. | I will leave it to you to reflect on this matter. | I will therefore leave it to you to consider this matter. |
> | Unfortunately, this hope has not become reality. | Unfortunately, this has not become reality. | Despite that, it has not become reality. | The reality is, however, rather different. |
> | Please could you take suitable action here. | Could you take suitable action? | Can you please do something about it? | Could you please take some action? |
> | There is no doubt that the directive will have to be revised if its objectives for the future are to be achieved. | There is no doubt that the directive will have to be revised if its objectives are to be met. | There is no room for fudge or doubt that the directive will need to be revised if it is to be implemented. | There is no doubt that the directive will have to be revised if its objectives are to be met. |
> | I am in favour of Turkey’s admittance to the EU, but the Copenhagen criteria must be met. | I am in favour of Turkey’s’s admittance to the EU, but the Copenhagen criteria must be met. | I am very much in favour of Turkey’s admittance to the EU, but these must be taken into account. | I am very much in favour of Turkey’s accession to the European Union, but the Copenhagen criteria must be met. |
>
> Overall, we observe that the baseline Self-Supervised approach tends to be more conservative in its transformation of the source input (tr) compared to the Joint Training approach. With Joint Training, not only do we witness changes in lexical choices, but also see alterations in sentence structure that retain the original meaning. For instance, Joint Training[1]-hTr exhibits a  more liberal transformation of the sentence "Unfortunately , this hope has not become reality ." to "The reality is, however , rather different .", capturing and conveying the meaning of the source input from a distinct perspective. Similarly, the last example highlights the added fluency from Joint Training[1]-hTr in the source input due to the expansion of the abbreviation EU and use of a more refined word choice for "admittance".
>
> **Q1. For the self-supervised part (Section 3.1), you use the SPE module to pick matching sentences for tr-og pairs that will be used to train the encoder-decoder. Each sentence is tested against every other sentence? What is this SPE module, more concretely?  Is it a BERT-like encoder? Did you use the one from (Ruiter et al., 2019) or did you train your own? On what corpora was SPE trained on?**
>
> Yes, the SPE module is used to extract (near-)parallel pairs from the comparable corpus to train the Encoder-Decoder.  Each sentence from Original (OG) is tested against all sentences in Translated (TR). We use FAISS index ([Johnson, et. al (2019)](https://arxiv.org/abs/1702.08734)) for this purpose, to make the search more efficient. Following [Ruiter et. al (2022)](https://aclanthology.org/2022.socialnlp-1.2), we build separate FAISS indexes for both TR and OG sentence representations and perform a bidirectional search to retrieve only high-quality pairs. This implies that during search time, the TR index is queried with an OG sentence vector to retrieve its nearest neighbors among the TR sentences, and vice-versa. This procedure results in two kinds of mappings: TR_to_OG and OG_to_TR. Only if a sentence-pair attains the highest rank in both directions, it is considered a candidate pair for further filtering, as described in Section 3.1 and in [Ruiter et. al (2019)](https://aclanthology.org/P19-1178).
>
> SPE is not quite a BERT encoder but a similarity matching module that receives translated and original latent representations from the Style Encoder to find and extract sentences with similar meanings ([Ruiter et. al (2019)](https://aclanthology.org/P19-1178)). Specifically, the module employs two types of representations, a) the sum of static word embeddings w(*) and b) the sum of contextualized encoder outputs e(*).  The embedded pairs {w(tr), w(og)} and {e(tr), e(og)} are individually scored using FAISS and a margin-based measure ([Artetxe  et. al (2018)]), and the top candidate pairs are selected for further filtering (as explained in Section 3.1).
> SPE is trained on the same comparable data (MPDE)  that is used for Style Transfer, and the two (SPE and Style Transfer) feed and improve each other in a virtuous loop. In SPE, parallel pairs are extracted in an online fashion. In every iteration, the Style Encoder learns to distinguish and represent the two styles (TR and OG) better. Consequently, the SPE module receives updated representations for TR and OG sentences in each iteration to be able to extract better quality pairs as the training progresses.
>
> **Q2. The SPE with threshold uses a threshold of 1.01. Can you say more on how this value was determined?**
>
> The value was determined empirically by inspecting values in the range of [0.95, 1.03], and validated through manual assessment of the quality of accepted pairs. We observed that with a higher threshold, fewer yet better quality original-translated parallel pairs are extracted. A threshold of 1.01 provided a good trade-off between quality and quantity.
>
> **Q3. When calculating the probability vectors, a temperature value of 0.1 is used. Can you say more on how this value was picked?**
>
> We are glad that you asked because we missed a citation. The value was taken following the work ([Unanue et. al (2021)](https://aclanthology.org/2021.acl-short.115)) to enforce sparsity.
>
> > Unsupervised approaches seem almost magical, getting results from apparently nothing.
>
> **(Soundness/Excitement)** Previous approaches use un- ([Lample et al. 2018](https://arxiv.org/abs/1711.00043), [Artetxe et al. 2019](https://aclanthology.org/P19-1309)) or self-supervised ([Ruiter et al. 2019](https://aclanthology.org/P19-1178), [Liu et. al. 2022](https://arxiv.org/pdf/2204.08123.pdf)) methods to address the lack of parallel training data. Here we use them to address the logical impossibility of mono-lingual translated - original parallel training and validation data. This may well be a first.

---

### Official Review · Reviewer_k8r3 · 2023-08-01

**Typos Grammar Style And Presentation Improvements:** 420
**Soundness:** 4

**Excitement:**

4: Strong: This paper deepens the understanding of some phenomenon or lowers the barriers to an existing research direction.

**Paper Topic And Main Contributions:**

The paper describes a method to modify ‘translationese’ so that they look like regular sentences originally written in the target language. This has important implications in the training and evaluation of different natural language processing models. The proposed method involves training a Transformer encoder-decoder model that performs the conversion. It is trained in an iterative way from a comparable corpus of translationese/non-translationese sentences in the same language with two auxiliary losses: one that enforces that the generated sentence has the style of a sentence originally written in the target language, and another one that enforces semantics preservation. The proposed approach is evaluated according to the following factors: whether an automatic classifier is able to distinguish between the outputs and ”original” text, content preservation, and perplexity according to a language model trained on “original” text. The proposed approach is better than a simpler baseline without the auxiliary losses according to the classifier, but content preservation and perplexity show a mixed picture.

The main contributions of the paper are:
- Presenting a new algorithm for changing the style of ‘translationese’ to “regular sentences”
- Being the first algorithm for this purpose that actually generates surface forms and does not require an Abstract Meaning Representation parser

**Questions For The Authors:**

A: If only one half of the test set changes between the different models being evaluated, why didn’t you report only results on that half? It would be clearer if you reported how many of the sentences of that half are considered by the classifier as “og”.

**Reasons To Accept:**

- The paper advances state of the art when converting translationese into regular sentences (as if they had been written from scratch instead of being translated)
- The paper is clear and well-written

**Reasons To Reject:**

The use of a classifier for determining how well the approach converts the translationese into regular sentences undermines the reliability of the conclusions, as changes in its accuracy can be attributed to the classifier’s own biases, in addition to the properties of the sentences generated by the system. A downstream evaluation of the output of the system, for instance, by training an NLI model as did by Dutta Chowdhury et al. (2022), would assess in a more precise way the performance of the proposed approach.

**Reproducibility:**

4: Could mostly reproduce the results, but there may be some variation because of sample variance or minor variations in their interpretation of the protocol or method.

**Reviewer Confidence:**

2: Willing to defend my evaluation, but it is fairly likely that I missed some details, didn't understand some central points, or can't be sure about the novelty of the work.

---

> ### Author Rebuttal · Authors · 2023-08-28
>
> We thank Reviewer 2 for the thorough and insightful feedback.
>
> > The use of a classifier for determining how well the approach converts the translationese into regular sentences undermines the reliability of the conclusions, as changes in its accuracy can be attributed to the classifier’s own biases, in addition to the properties of the sentences generated by the system.
>
> **(Soundness of using a classifier)** Previous research has shown that human experts (professional translators) are usually not able to reliably distinguish between originally authored and professionally translated texts in the same language, genre, topic etc. beyond chance ([Tirkkonen-Condit et. al (2002)](https://api.semanticscholar.org/CorpusID:144823252), [De Sutter et. al (2020)](https://www.tandfonline.com/doi/abs/10.1080/0907676X.2019.1611891)).  Reliably detecting translationese requires either (i) careful corpus analysis of sufficiently large volumes of original and translated data ([Gellerstam et. al (1986)](https://api.semanticscholar.org/CorpusID:59685951), [Baker et. al(1993)](https://api.semanticscholar.org/CorpusID:57174748), [Laviosa et. al (1998)](https://api.semanticscholar.org/CorpusID:678873)
> , [Koppel et. al (2011)](https://aclanthology.org/P11-1132)) to show differences in the distribution of e.g. discourse connectives and/or other translationese manifestations, or (ii) the use of classifiers trained on original and translationese data ([Baroni et. al (2006](ttps://api.semanticscholar.org/CorpusID:11006810), [Kurokawa et. al (2009)](https://aclanthology.org/2009.mtsummit-papers.9), [Rabinovich et. al (2015)](https://aclanthology.org/Q15-1030), [Volansky et. al (2015)](https://api.semanticscholar.org/CorpusID:5133943), [Pylypenko et al. (2021)](https://aclanthology.org/2021.emnlp-main.676)]). Given enough training data and given enough data at prediction time, translationese classifiers can be astonishingly accurate, with up to > 90% accuracy ([Rabinovich et. al (2015)](https://aclanthology.org/Q15-1030),  [Pylypenko et al. (2021)](https://aclanthology.org/2021.emnlp-main.676), [Amponsah et al. (2022)](https://aclanthology.org/2022.blackboxnlp-1.23)]). In our work, we use SOTA BERT-based approaches as e.g. by [Pylypenko et al. (2021)](https://aclanthology.org/2021.emnlp-main.676). Given that the MPDE EuroParl data ([Amponsah et. al (2021)](https://aclanthology.org/2021.motra-1.1)) we use is fairly uniform in terms of genre and domain, we (i) expect the classifiers to be reliable and (ii) reduced classifier accuracy to (near) random (per classification datum/input) to be a strong indication that (any previous) translationese signals have been reduced in the data under classification. In this respect, the methodology and approach we follow is as good and as bad (as sound) as the current SOTA in translationese research at large ([Pylypenko et al. (2021)](https://aclanthology.org/2021.emnlp-main.676), [Dutta Chowdhury et al. (2022)](https://aclanthology.org/2022.naacl-main.292)). That said, we do take the point raised by the reviewer: [Amponsah et al. (2022)](https://aclanthology.org/2022.blackboxnlp-1.23) show some evidence that high-performance translationese classifiers may use spurious correlations in the data (such as topic) that are not translationese signals. We are aware of previous research e.g. [Dutta Chowdhury et al.(2020)](https://aclanthology.org/2020.coling-main.532) that uses masking (e.g. PoS) to abstract from text data to mitigate possible topic influences in translationese data. However, they work at the level of word embedding spaces (hidden representations), while we are focusing on generating surface text (converting translationese into original style text). Quantifying (and perhaps even reducing) spurious correlations in (translationese) classification remains an important topic not addressed in our paper. If the paper is accepted, we will use the extra page to discuss the point raised by the reviewer following the outline above.
>
> > A downstream evaluation of the output of the system, for instance, by training an NLI model as did by Dutta Chowdhury et al. (2022), would assess in a more precise way the performance of the proposed approach.
>
> We fully agree: extrinsic evaluation, such as NLI as in [Dutta Chowdhury et al. (2022)](https://aclanthology.org/2022.naacl-main.292) would be very useful to assess the performance of our approach. We looked at the data. Unfortunately, the substantial topic differences between the NLI data and our MPDE EuroParl data ([Amponsah et. al (2021)](https://aclanthology.org/2021.motra-1.1)) require retraining our translating away translationese system to fit it to the NLI data. Without retraining, our approach would import unwanted domain differences over and above translationese mitigation in the conversion. This is related to question W1 of the reviewer above. We will explore our approach on the NLI data as part of our future work.
>
> **Q1: If only one half of the test set changes between the different models being evaluated, why didn’t you report only results on that half? It would be clearer if you reported how many of the sentences of that half are considered by the classifier as “og”.**
>
> **(Soundness):** Classification accuracies on only style-transferred human-translation (tr) outputs are shown below (in Table A) and are in line with the results on all inputs (tr + og) reported in the paper (compare Table 2). Lower accuracy on style-transferred tr, i.e. “mis-classifying” them as og, indicates that style transfer successfully mitigates translationese.
>
> *Table A: Classification results on 10,305 human translation (tr) test sentences of EN-ALL MPDE*
> |   EN-ALL    | Accuracy | # og-classified sentences  |
> |         :---:        |    :----:   |          :---: |
> | Human-Translation (tr) half     | 82.38       | 1816   |
> | Self-Supervised[1]   | 66.72        | 3430     |
> | Self-Supervised[2]   | 58.87        | 4239     |
> | JointTraining[1]-hTr   | 32.00        | 7008    |
> | JointTraining[2]-hTr  | 29.13         | 7303    |
> | JointTraining[1]-mTr   | 32.74        | 6931    |
> | JointTraining[2]-mTr  | 30.24         | 7189    |
>
> > The proposed approach is better than a simpler baseline without the auxiliary losses according to the classifier, but content preservation and perplexity show a mixed picture.
>
> **(Soundness)** Regarding the mixed results in content preservation, it is important to note that we compute BERTScore between the input-translated sentences and the style-transferred outputs. Therefore, if more sentences in the output are identical to the input, BERTScore would be higher. The scores in Tables 2 and 3 indicate that the self-supervised approach yields a higher number of identical sentences compared to the Joint Training method, and therefore, achieves a higher BERTScore indicating less Translationese to Original style-transfer. In a sense, BERTScore is sensitive to both meaning and surface changes ([Hanna et. al. (2021)](https://aclanthology.org/2021.wmt-1.59), [Zhou et. al. (2022)](https://aclanthology.org/2022.acl-short.45)).
> Regarding perplexity, the main goal is to assess whether the perplexities from both approaches are lower than those on the Human Translated test set when using a language model trained on originals. Perplexities in Tables 2 and 3 in the paper do not show a consistent trend across the languages in the two setups and we agree that this prompts at least the need for further manual inspection of the style-transferred outputs. We provide a short comparison in Table  B and will extend this in the revised version of the paper.
> Additionally, we think our response to Reviewer 3 about the manual analysis of examples (Table C) can also be of interest in the context of the concern raised here.
>
> *Table B: Comparison of the outputs from the Self-Supervised and Joint Training system, using the same translated input sentences.*
>
> | Source (tr) | Hypothesis1: Self-Supervised[2] | Hypothesis2: JointTraining[1]-hTr |
> | :---: | :---: | :---: |
> | That is the first issue | That is the first issue | That is the first issue |
> | That is why I am asking for your support. | That is why I am asking for your support. | That is why I am asking for your assistance. |
> | Please could you take suitable action here . | Could you take suitable action ? | Could you please take some action ? |
> | They have presented their opinions very well , and I thank them for this . | They have done so very well , and I thank them for that . | They have expressed their views on this subject and I thank them for that . |
>
> Table C presents the style-transferred outputs resulting from both the baseline Self-Supervised approach and the proposed Joint Training approach. These outputs are generated using the same translated source sentence (tr) as input. Depending on the source sentence, both models either yield consistent outputs (the first row), or they exhibit differences in the lexical choice and sentence structure while still effectively capturing the meaning of the input (rows 2-3). In the last example, we observe that the baseline approach opts for a more general description (“done so”), potentially leading to a somewhat less specific representation. On the other hand, the Joint Training approach is more descriptive with its use of “expressed their views on this subject”, enhancing the overall clarity and concreteness of the conveyed source sentence.

---

### Official Review · Reviewer_PG8x · 2023-08-05

**Soundness:** 4
**Typos Grammar Style And Presentation Improvements:** Line number 533 "translation. to rThis"

**Excitement:**

4: Strong: This paper deepens the understanding of some phenomenon or lowers the barriers to an existing research direction.

**Paper Topic And Main Contributions:**

The paper explores a novel approach to reduce translationese in translated texts which is a translation-based style transfer. As there is no parallel translated and original data in the same language, the author uses a self-supervised approach that can learn from comparable mono-lingual original and translated data.

**Reasons To Accept:**

The paper is well written, and experiments are described well.
The authors not only explain the challenges and provide solutions to the challenges but also provide a suitable justification for the solution.
The method combines self-supervised and unsupervised learning, eliminating the need for parallel corpora, which doesn't exist for this particular task. Moreover, the method extended to low-resource settings as well.


**Reasons To Reject:**

None

**Reproducibility:**

4: Could mostly reproduce the results, but there may be some variation because of sample variance or minor variations in their interpretation of the protocol or method.

**Reviewer Confidence:**

4: Quite sure. I tried to check the important points carefully. It's unlikely, though conceivable, that I missed something that should affect my ratings.

---

> ### Author Rebuttal · Authors · 2023-08-28
>
> We thank Reviewer 1 for the thoughtful feedback. We address  translationese mitigation through translation-based style transfer. Due to the impossibility of obtaining parallel mono-lingual translationese-original data, our approach centers on a self-supervised NMT approach that simultaneously learns to perform two tasks in an online fashion: 1) extract (near) parallel pairs from a comparable original-translationese corpus, and 2) perform style transfer using the parallel pairs (a task made complex by the absence of parallel translated and original data in the same language). To eliminate the need for parallel validation data and to effectively utilize the monostylistic translationese data, we extend the baseline self-supervised system with unsupervised learning by combining a language model loss and a semantic similarity loss. Our experimental results indicate our approach is able to reduce translationese classifier accuracy to a level of a random classifier after style transfer while adequately preserving the content and fluency in the target original style.

---

### Meta-Review · Area_Chair_7HnN · 2023-09-18

**Recommendation:** 5

**Metareview:**

The reviewers have praised this papers' technical contribution to the field and originality. They agree that the paper advances the state-of-the-art in machine translation particularly concerning the well-studied phenomenon of translationese.

Finally, the reviewers also praised the paper style, writing, and clarity.

---

### Decision · Program_Chairs · 2023-10-07

**Decision:**

Accept-Main

**Comment:**

The reviewers have praised this papers' technical contribution to the field and originality. They agree that the paper advances the state-of-the-art in machine translation particularly concerning the well-studied phenomenon of translationese.

Finally, the reviewers also praised the paper style, writing, and clarity.